# The Advanced Toolbox for Multitask Medical Imaging Consistency (ATOMMIC): A framework to facilitate Deep Learning in Magnetic Resonance Imaging

**Dimitrios Karkalousos**[1,2,3]                 D.KARKALOUSOS@AMSTERDAMUMC.NL
[1] *Department of Biomedical Engineering & Physics, Amsterdam University Medical Center, Location University of Amsterdam, Amsterdam, The Netherlands*
[2] *Department of Radiology & Nuclear Medicine, Amsterdam University Medical Center, Location University of Amsterdam, Amsterdam, The Netherlands*
[3] *Informatics Institute, University of Amsterdam, Amsterdam, The Netherlands*

**Ivana Išgum**[1,2,3]                        I.ISGUM@AMSTERDAMUMC.NL
**Henk A. Marquering**[1,2,4]               H.A.MARQUERING@AMSTERDAMUMC.NL
[4] *Amsterdam Neuroscience, Brain Imaging, Amsterdam, The Netherlands*

**Matthan W.A. Caan**[1,4]                   M.W.A.CAAN@AMSTERDAMUMC.NL

**Editors:** Accepted for publication at MIDL 2024

## Abstract

Integrating Deep Learning (DL) into medical imaging, particularly in Magnetic Resonance Imaging (MRI), has marked a significant advancement in the field, enhancing the efficiency and accuracy of tasks such as image reconstruction, segmentation, and quantitative parameter map estimation. Despite these advancements, existing frameworks have limited support to perform multiple tasks simultaneously, essential for optimizing the workflow from data acquisition to analysis. Addressing this gap, we introduce the Advanced Toolbox for Multitask Medical Imaging Consistency (ATOMMIC), a novel open-source toolbox designed to facilitate the integration of multiple MRI tasks within a unified MultiTask Learning (MTL) framework. ATOMMIC supports a wide range of DL models and datasets, allowing for seamless and consistent execution of multiple tasks. By enabling joint task execution and supporting complex and real-valued data, ATOMMIC allows to streamline various DL applications in MRI reconstruction and analysis.

**Keywords:** Deep Learning, Magnetic Resonance Imaging, Image reconstruction, Image segmentation, Multitask Learning

## 1. Introduction

The availability of large public datasets and sophisticated frameworks has facilitated the rapid growth of Artificial Intelligence (AI) applications in medical imaging. Deep Learning (DL) models can significantly accelerate the acquisition of Magnetic Resonance Imaging (MRI) while improving the quality of reconstruction. DL techniques have also been extended to accurate and precise image segmentation and more complex tasks, such as end-to-end quantitative parameter map estimation in MRI. These are pivotal steps toward the clinical objective of segmenting or classifying a disease's anatomy and pathology. Existing AI frameworks for medical imaging often limit researchers on performing tasks independently; e.g., the reconstruction and segmentation tasks, although related, meaning that the

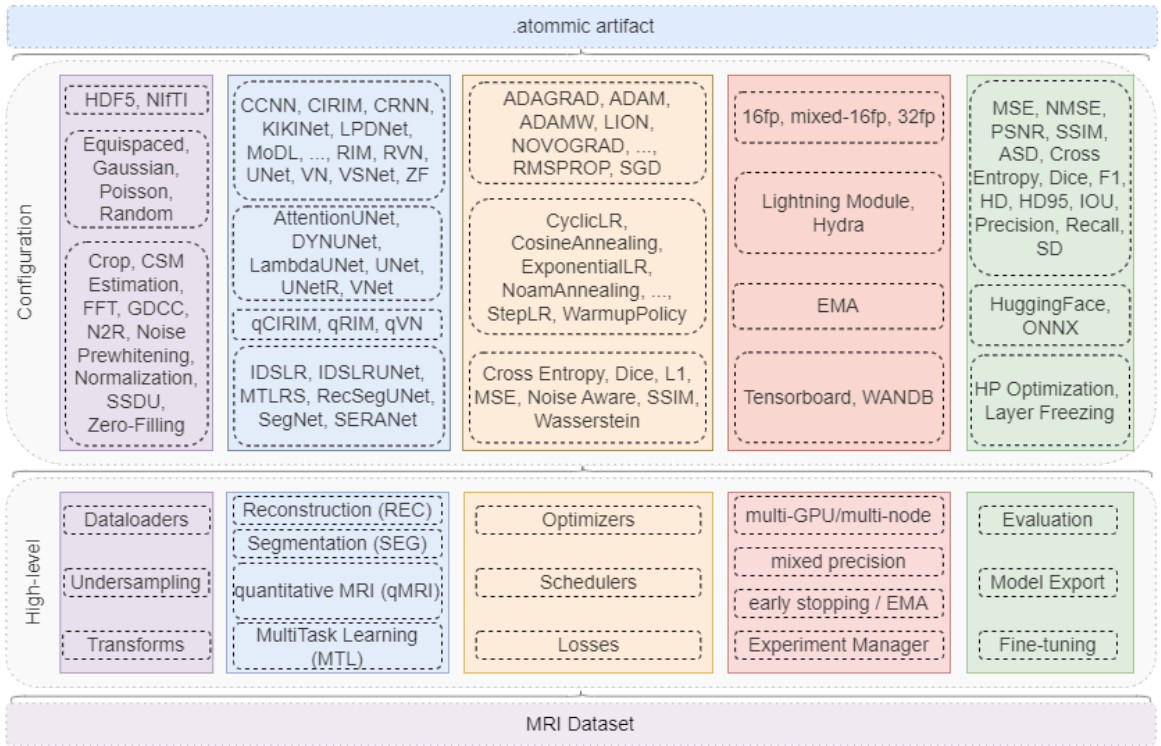

Figure 1: Schematic overview of ATOMMIC. An `MRI dataset` is given as input (bottom-first tier). Next, `High-level` configurations are defined (second tier), such as undersampling configurations, transforms, task(s), optimizers, schedulers, losses, trainer, and export. In the `Configuration` (third tier), we can define the exact methods and parameters for training, logging, and versioning through a YAML file. The output is the `.atommic artifact` (fourth tier), containing the model checkpoints and configuration, which can be directly downloaded and used for testing.

performance of a segmentation model will depend on the reconstruction quality of the input image, cannot be performed simultaneously. The Medical Open Network for Artificial Intelligence (MONAI) ((Cardoso et al., 2022)) is a widely used research framework that integrates multiple medical imaging tasks, imaging modalities, and data types. However, tasks can only be performed independently, and complex-valued data support is limited to the reconstruction task. End-to-end AI solutions aim to build a multitask model that moves from the acquired data directly to the objective. Here, intermediate tasks can be integrated to improve performance and efficiency by minimizing the time overhead associated with the execution of independent tasks (Adler et al., 2022). The challenges lie in the variations in data structures, formats, and programming languages, which underscores the need for comprehensive AI solutions in medical image analysis.

We propose the Advanced Toolbox for Multitask Medical Imaging Consistency (ATOMMIC), an open-source versatile toolbox integrating various MRI tasks, such as image reconstruction, image segmentation, and quantitative parameter map estimation. Using Mul-

tiTask Learning (MTL), related tasks, such as reconstruction and segmentation, can be performed jointly. The toolbox ensures consistency across models, supported datasets, and training and testing methodologies, accommodating complex and real data formats. Training and testing DL models is straightforward using a single configuration YAML file with options on MRI transforms, undersampling, and model hyperparameters, as illustrated in Fig. 1. Nine publicly available datasets are supported, with complex and real-valued data and twenty-five DL models. A detailed list of models and datasets is available in the Appendix (Table 1).

ATOMMIC is built on top of NVIDIA's NeMo ((Kuchaiev et al., 2019)), a computationally efficient conversational AI toolkit that allows high-performance training and testing with multiple GPUs, nodes, and mixed precision support. The code is available on GitHub[1] under the Apache 2.0 license, fostering transparency, reproducibility, and collaborative research in medical imaging.

## 2. Discussion

The Advanced Toolbox for Multitask Medical Imaging Consistency (ATOMMIC) is a valuable Deep Learning (DL) framework applied to various MRI tasks, including image reconstruction, image segmentation, and quantitative parameter map estimation. It uses MultiTask Learning (MTL) for joint reconstruction and segmentation. Consistency in task performance is ensured by harmonizing network implementations, hyperparameters, image transformations, and training configurations, accommodating complex and real-valued data support. We aim to empower the research community with a multitask DL framework facilitating MR imaging for different tasks, allowing model sharing and standardized pre-processing pipelines for public datasets. ATOMMIC can be expanded through the community to include essential tasks such as classification, registration, and motion correction, ultimately creating a comprehensive end-to-end multitask framework that simplifies medical image reconstruction and analysis.

## 3. Acknowledgments

This publication is based on the STAIRS project under the TKI-PPP program. The collaboration project is co-funded by the PPP Allowance made available by Health Holland, Top Sector Life Sciences & Health, to stimulate public-private partnerships.

H.A. Marquering and M.W.A. Caan are shareholders of Nicolab International Ltd. H.A. Marquering is a shareholder of TrianecT B.V. and inSteps B.V. (unrelated to this project; all paid individually).

---

1. https://github.com/wdika/atommic

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

## Appendix

Table 1: Overview of natively supported tasks, models, and datasets in ATOMMIC. The first column reports the supported tasks, which are MultiTask Learning (MTL), quantitative MRI (qMRI), Reconstruction (REC), and Segmentation (SEG). The second column reports the supported models, and the third column reports the publicly available datasets supported.

| Task | Models | Datasets |
|------|--------|----------|
| MTL | Image domain Deep Structured Low-Rank Network (IDSLR) (Pramanik and Jacob, 2021) Image domain Deep Structured Low-Rank UNet (IDSLRUNet) (Pramanik and Jacob, 2021) Multi-Task Learning for MRI Reconstruction and Segmentation (MTLRS) (Karkalousos et al., 2024) Segmentation Network MRI (SegNet) (Sun et al., 2019) | SKM-TEA (Desai et al., 2022) |
| qMRI | quantitative Cascades of Independently Recurrent Inference Machines (qCIRIM) (Zhang et al., 2022) quantitative End-to-End Variational Network (qVarNet) (Zhang et al., 2022) quantitative Recurrent Inference Machines (qRIM) (Zhang et al., 2022) | AHEAD (Alkemade et al., 2020) |

| | | |
|---|---|---|
| REC | Cascades of Independently Recurrent Inference Machines (CIRIM) (Karkalousos et al., 2022) Convolutional Recurrent Neural Networks (CRNNet) (Qin et al., 2019) Deep Cascade of Convolutional Neural Networks (CascadeNet) (Schlemper et al., 2018) End-to-End Variational Network (VarNet) (Sriram et al., 2020) Independently Recurrent Inference Machines (IRIM) (Karkalousos et al., 2022) Joint Deep Model-Based MR Image and Coil Sensitivity Reconstruction Network (JointICNet) (Jun et al., 2021) KIKINet (Eo et al., 2018) Learned Primal-Dual Net (LPDNet) (Adler and Oktem, 2018) Model-based Deep Learning Reconstruction (MoDL) (Aggarwal et al., 2019) Recurrent Inference Machines (RIM) (Lønning et al., 2019) Recurrent Variational Network (RVN) (Yiasemis et al., 2022) UNet (Ronneberger et al., 2015) Variable Splitting Network (VSNet) (Duan et al., 2019) XPDNet (Ramzi et al., 2020) Zero-Filled reconstruction (ZF) (Pruessmann et al., 1999) | AHEAD (Alkemade et al., 2020) CC359 (Beauferris et al., 2022) fastMRI Brains Multicoil (Zbontar et al., 2019) fastMRI Knees Multicoil (Zbontar et al., 2019) fastMRI Knees Singlecoil (Zbontar et al., 2019) SKM-TEA (Desai et al., 2022) Stanford Knees (Epperson et al.) |
| SEG | AttentionUNet (Oktay et al., 2018) DYNUNet (Isensee et al., 2021) UNet 2D & 3D (Ronneberger et al., 2015) VNet (Milletari et al., 2016) | BraTS 2023 Adult Glioma (Kazerooni et al., 2024) ISLES 2022 Sub Acute Stroke (Hernandez Petzsche et al., 2022) SKM-TEA (Desai et al., 2022) |

