# OpenReview forum: "The Advanced Toolbox for Multitask Medical Imaging Consistency (ATOMMIC): A framework to facilitate Deep Learning in Magnetic Resonance Imaging"
_MIDL.io/2024/Short_Papers — MIDL 2024 Short Papers_

### Official Review · Reviewer_ywpo · 2024-04-24

**Confidence:** 5
**Final Rating:** 4

**Review:**

This is a very well written short paper presenting ATOMMIC, a framework for training deep, multi-task ML models on MRI datasets. ATOMMIC builds on NeMo and is more flexible than previous frameworks, particularly in the multi-tasking aspect (i.e., one can train models for simultaneous segmentation and super-resolution).

Even though there's no new methodology or results in the article, I think that the MIDL audience will be very interested hearing about this framework.

One suggestion for the authors: maybe replace "facilitate MR imaging" in the title by "facilitate MR image analysis" or even "blah blah (ATOMMIC): a framework to facilitate deep learning in MRI"

---

### Decision · Program_Chairs · 2024-04-26

Accept